# A Small-Scale Analysis of Elevational Species Richness and Beta Diversity Patterns of Arthropods on an Oceanic Island (Terceira, Azores)

**DOI:** 10.3390/insects12100936

**Published:** 2021-10-14

**Authors:** Jan Peter Reinier de Vries, Emiel van Loon, Paulo A. V. Borges

**Affiliations:** 1cE3c—Centre for Ecology, Evolution and Environmental Changes, Azorean Biodiversity Group, Faculdade de Ciências Agrárias e do Ambiente, Universidade dos Açores, 9700-042 Angra do Heroísmo, Açores, Portugal; vries.reinier@gmail.com; 2IBED—Institute of Biodiversity and Ecosystem Dynamics, University of Amsterdam, 1012 WX Amsterdam, The Netherlands; E.E.vanLoon@uva.nl

**Keywords:** insects, altitude, biodiversity, alpha-diversity, replacement, disturbance

## Abstract

**Simple Summary:**

We studied the diversity of arthropods in native forests along a 1000 m elevation gradient on Terceira Island, Azores (Portugal). These forests form an isolated and threatened habitat with unique endemic species. We analysed the change in alpha and beta diversity of arthropod species with elevation and if the diversity of endemic, native non-endemic and introduced species responds differently to elevation. Resident arthropods were sampled using SLAM (Sea, Land and Air Malaise) traps between 2014 and 2018. Spiders (Araneae), beetles (Coleoptera), true bugs (Hemiptera) and barklice (Psocoptera), as well as endemic, native and introduced species, were analysed separately. Total species richness decreases with elevation for all species, Coleoptera and Psocoptera, and particularly so for introduced species, but peaks at mid-high elevation for Araneae and endemic species. These patterns are probably driven by unfavourable climatic conditions at higher elevations while being influenced by human disturbance at lower elevations. Total species diversity along the whole elevation gradient is shaped by this decreasing richness as well as the replacement of species at different elevations.

**Abstract:**

We present an analysis of arthropod diversity patterns in native forest communities along the small elevation gradient (0–1021 m a.s.l.) of Terceira island, Azores (Portugal). We analysed (1) how the alpha diversity of Azorean arthropods responds to increasing elevation and (2) differs between endemic, native non-endemic and introduced (alien) species, and (3) the contributions of species replacement and richness difference to beta diversity. Arthropods were sampled using SLAM traps between 2014 and 2018. We analysed species richness indicators, the Hill series and beta diversity partitioning (species replacement and species richness differences). Selected orders (Araneae, Coleoptera, Hemiptera and Psocoptera) and endemic, native non-endemic and introduced species were analysed separately. Total species richness shows a monotonic decrease with elevation for all species and Coleoptera and Psocoptera, but peaks at mid-high elevation for Araneae and endemic species. Introduced species richness decreases strongly with elevation especially. These patterns are most likely driven by climatic factors but also influenced by human disturbance. Beta diversity is, for most groups, the main component of total (gamma) diversity along the gradient but shows no relation with elevation. It results from a combined effect of richness decrease with elevation and species replacement in groups with many narrow-ranged species.

## 1. Introduction

Biodiversity is a complex concept that is most often assessed by analysing taxonomic species richness at different spatial scales [1,2]. Ecological processes shaping taxonomic diversity can be addressed through alpha-, beta- and gamma-diversity patterns, as first proposed by Whittaker using a multiplicative approach to decompose gamma diversity (gamma = alpha × beta) [3]. However, Lande [4] suggested an additive partition proposing that gamma = alpha + beta. This meant that beta diversity could be measured in terms of species numbers in the same way as alpha and gamma, instead of being the unitless measure of turnover resulting from multiplicative partition. This framework addresses the effects of spatial scale, which is an essential factor in diversity studies [5,6,7]. Beta diversity can be partitioned into two distinct components that provide crucial insights into the processes generating diversity: species loss (or gain), driven by physiological constraints, and species replacement (or turnover), driven by specialisation and thus related to small ranges and high endemism [8]. Alternatively, beta diversity can be partitioned into turnover and nestedness-resultant dissimilarity/richness difference components [9].

How biodiversity changes along environmental gradients is a key research topic in ecology and biogeography [1,6,10,11]. Elevation influences multiple environmental factors, creating a great variation in conditions and habitats on a short spatial range [12]. Therefore, elevation is a main driver of diversification and endemism [10,13,14] and may relatively be an especially crucial driver of speciation in temperate regions [13]. Since multiple gradients with differing conditions can be found all over the world, elevation gradients offer optimal conditions to study the patterns and driving factors of species distributions, to provide insight into fundamental drivers of biodiversity [10,12,13]. Moreover, elevation gradients can provide insight into the effects of human disturbances that often change along gradients, and they provide a baseline record for future studies on, e.g., global change effects and conservation [1,6,15,16,17,18]. Their clear boundaries and isolation, enhancing in situ speciation and limiting the mixing of species pools, make oceanic island gradients especially valuable study sites [10,19,20,21,22].

Multiple studies have shown that the effect of elevation on species richness varies between locations and taxonomical groups but tends to fall into four distinct patterns: monotonic decreases; low-elevation plateaus followed by decreases or mid-elevation peaks; and unimodal mid-elevation peaks [17], whereas in rare cases increases occur [6]. Mid-elevation peaks appear to be the dominant pattern, as found by Rahbek [5] among 204 studies of mainly plant communities; in meta-analyses and multiple-site studies on small mammals [1,23], ferns [24], ants [18,25,26] and moths [27]; as well as in multiple single gradient studies, e.g., of bryophytes on the Macaronesian islands of La Palma [7] and Terceira [28] and insects on Hawaii island [29].

Monotonic decreases are the second-most common pattern, with high frequencies in meta-analyses on birds, bats and reptiles [17,30,31]. Decreasing patterns are also commonly found in arthropod studies [6,15,32], e.g., of ants in the Great Smoky Mountains, USA [33], bees on Mt. Kilimanjaro, Tanzania [34], butterflies on Mt. Olympus, Greece [35] and beetles in the Genting Highlands, Malaysia [36].

Despite their abundance and ecological importance, elevational diversity patterns of arthropods remain relatively poorly documented and understood, especially on oceanic islands. Furthermore, compositional patterns (beta diversity) remain relatively poorly studied [1,37,38]. Replacement is often the dominant beta-diversity component and is commonly assumed to be a main determinant of species richness, at least on large spatial scales [1,39]. Several studies found replacement peaks at major ecotones [1,37,40]. Large-scale studies on plants [38] and vertebrates [39] however found no consistent replacement patterns, but instead a high variability and poor correlation with species richness. McCain and Beck [39] concluded that species range distributions are individualistic and largely stochastic in nature.

The explanation of diversity patterns remains challenging and is, in most cases, driven by interactions between various biogeographical processes rather than independent dominant factors [1,6,18]. Environmental factors, notably temperature and humidity, are often regarded as the most important drivers and induce group- and site-specific diversity responses [6,17,24,30]. Strong support for spatial hypotheses, such as species-area and mid-domain effects, has been found in several studies but cannot explain the variation in diversity patterns among both locations and taxa [1,6,17,25].

Oceanic islands generally host a rich and to a large proportion unique biodiversity [41] but are especially sensitive to external influences. Therefore, oceanic islands are disproportionally affected by various anthropogenic pressures, causing habitat loss and species extinction [42,43,44]. These factors highlight the importance of biodiversity studies on oceanic islands [21,41], particularly in their remaining natural habitats, of which native forests are amongst the most substantial and diverse [19,22,44]. For example, the Azores, an isolated archipelago in the North Atlantic ocean, harbours rare and vulnerable forests that have been severely affected by human impacts [45]. Its third-largest island, Terceira, hosts the largest and most pristine remains of the formerly widespread Azorean native forests, that are of critical importance for the survival of many highly threatened and endemic taxa [46]. Native forest loss has severely impeded the archipelago’s native biota and still poses a high extinction risk today [42].

Long-term monitoring is crucial to provide quantitative measures of ecological changes in such threatened island ecosystems, including effects of anthropogenic pressures, management actions and global changes (e.g., climate), as well as to study general ecological patterns [11,44,45]. On Terceira island, systematic monitoring of insect biota of the native forests has been conducted by permanent SLAM (Sea, Land and Air Malaise) traps since 2012, providing a continuous record of samples collected per three-month intervals [47,48,49]. Being situated along an elevation gradient from sea level to 1021 m a.s.l., this dataset can provide detailed insight into the effects of elevation on species diversity and abundance.

Here we present analyses of the relations between elevation and species diversity as well as elevation and abundance on Terceira island, testing the shares of different alpha- and beta-diversity components and comparing several arthropod subgroups. To further explore the contribution of spatial gradients to species richness formation, total beta diversity is partitioned into replacement and richness difference components [8].

We addressed three main questions: (1) How does the alpha diversity of Azorean arthropods respond to increasing elevation; (2) how do alpha diversity patterns across the elevation gradient differ between endemic, native non-endemic and introduced (alien) species; (3) what are the contributions of species replacement and species richness differences to beta diversity partitioning along the elevation gradient? Previous studies on the impact of land-use changes on the Azorean arthropod fauna have shown that endemic and native non-endemic species tend to be restricted to native vegetation [50,51,52,53,54,55], while introduced species favour non-native and disturbed vegetations [51,52]. The sampling sites at high elevation are situated in undisturbed native vegetation, but the sites at low- to mid-elevation, despite being dominated by native vegetation, are surrounded by a large matrix of non-native habitats. Based on these known patterns, we predict that: (1) Arthropod alpha diversity may be highest at mid-elevations, coinciding with the transition from disturbed vegetation with many introduced species to more pristine vegetation with many indigenous species. (2) The effect of elevation on species richness will differ between endemic/native non-endemic species and introduced species, since endemic and native non-endemic species richness may be highest at mid-to-high elevations but introduced species richness will be highest at low elevations. (3) High source–sink dynamics, with many introduced species coming from the surrounding anthropogenic habitats [48], result in higher beta-diversity rates for introduced species. Furthermore, given the high replacement levels of introduced species [48] and the disturbance differences along the gradient, as well as insights of other studies [39,48], we expect replacement (turnover) to be the most important component of beta diversity.

## 2. Materials and Methods

### 2.1. Study Area

Terceira is the Azores archipelago’s third-largest island, covering 402 km^2^ at an elevational range of 0–1021 m a.s.l. (Figure 1). Its climate is mild, with limited temperature fluctuation, a high humidity throughout the year, and a linear decrease in temperature but an increase in precipitation with elevation [2,56]. Terceira’s highland areas harbour the largest and most pristine remains of the Azores’ native forests, represented by the Terceira Natural Park, but its lowland areas have been extensively cultivated, and their former native forests have disappeared [28,42,46,50]. In total, 13 SLAM (Sea, Land, and Air Malaise) traps have been set up at permanent sites on Terceira island since 2012 and are still running (Figure 1). SLAM traps (110 cm × 110 cm × 110 cm) differ from the classical Malaise traps since they are designed in a way that flying insects are intercepted from four different directions (see Appendix A). Moreover, the SLAM also works as an extension of the soil, and animals can also walk into it instead of flying. Therefore, its monitoring range goes beyond the initially predicted high dispersive flying insect groups (Diptera, Hymenoptera, Lepidoptera). Trap 1 is situated in low-elevation woodland dominated by *Erica azorica*, traps 2–3 are situated in forest consisting of a mixture of indigenous and introduced species (*Pittosporum undulatum*), while the remaining traps 4–13 are situated in high-elevation natural forest dominated by endemic tree species (notably *Laurus azorica*, *Juniperus brevifolia* and *Ilex azorica*), the shrubs *Vaccinium cylindraceum* and *Myrsine retusa* and a high diversity of ferns. One of these traps (13) had persisting field problems and was therefore excluded from this study. From the remaining 12 sites, we selected a transect of five sites that represent the island’s elevation gradient at elevations of 46, 231, 404, 693 and 930 m a.s.l., respectively. These sites are situated at the Western side of the island, which is the only area where native forest vegetation is still present along the whole elevation gradient (Figure 1, see also 28). Second, to assess species diversity patterns over the whole island, we arranged the 12 sampling sites into five elevation bands of 0–200, 200–400, 400–600, 600–800 and 800–1000 m a.s.l., that represent one, one, two, six and two sites, respectively, at mean elevations of 46, 231, 490, 672 and 910 m a.s.l. The high number of sites above 600 m a.s.l. reflects the fact that the island’s remaining native forests are mostly restricted to high-elevation areas [46].

### 2.2. Data Collection

SLAM traps sampled both high-dispersive flying insects and terrestrial species that crawl into the traps on a seasonal basis. Samples were collected with roughly 90-day intervals, around the 15th of March (winter), June (spring), September (summer) and December (autumn). SLAM trap recipient was filled with propylene glycol (pure 1,2-PROPANODIOL) to kill and conserve the samples between periodic collections, safeguarding DNA preservation for genetic analysis. Borges et al. [47,49], Costa and Borges [58] and Matthews et al. [48] described the field procedures elaborately. We analysed the data from the period autumn 2014–summer 2018 when all sites were in operation. All samples were included in the analyses across all sites, but for the elevation transect, only summer samples were analysed to avoid seasonal fluctuations, representing four samples per site but only three for the highest site due to a field problem. The samples were sorted by order and, for the majority of orders, identified by stereo-microscope to taxonomical species or (in a minority of cases) morpho-species level. This identification procedure is considered to provide a robust basis for ecological analyses [18,47]. Specimens that could not be identified beyond order level, which includes the orders Acari, Amphipoda, Collembola, Diptera, Isopoda, Hymenoptera (except Formicidae) and Lepidoptera, were registered as such and are excluded in this study. Adults and juveniles were registered separately, and because juveniles include a large share of non-identified specimens, only adult records were analysed in this study. The counts of each sample were thereafter converted to a 90-day equivalent to correct for slight variations in sampling duration (it was impossible to visit all the sites on the same day). Data preparations were performed in R 3.5.3 [59].

### 2.3. Data Analyses

Analyses were performed for all identified species, belonging to the arthropod orders Araneae, Blattaria, Coleoptera, Hemiptera, Hymenoptera (Formicidae only), Julida, Lithobiomorpha, Microcoryphia, Neuroptera, Opiliones, Pseudoscorpiones, Psocoptera, Scutigeromorpha and Thysanoptera; hereafter referred to as ‘all species’ in this manuscript (Appendix A). Moreover, analyses were performed for seven subgroups; three endemic, native (but non-endemic) and introduced species [60], and four representing the orders Araneae, Coleoptera, Hemiptera and Psocoptera, which are sufficiently diverse and abundant across the elevation gradient to perform separate analyses. First, species elevational ranges were assessed based on the minimum, maximum and abundance-based mean elevation of each species’ records along the five-site elevation transect, assuming continuous ranges. Second, we examined alpha-diversity patterns. As the sampling methodology is considered to be highly standardised, and the abundances of the most dominant species were found to be highly different between elevations, the samples were not rarefied to a common number of individuals or a common coverage level. The five transect sites already sustain equal sampling effort (see above), but the dataset of all 12 sites was rarefied to a common sampling level of 15 samples per elevation band, using EstimateS analysis software [61] using 100 randomisations and default settings.

Sampling completeness was assessed by Species Accumulation Curves (SACs) and coverage estimates; the latter derived with the R package iNEXT [62]. Coverage levels were generally above 80% and were nearly 100% for all species, Hemiptera and native species, with slightly higher coverage levels obtained over all 12 sites than over the five transect sites (Appendix A). However, coverage levels were low for Coleoptera and introduced species at 930 m a.s.l. and Psocoptera at 693 m a.s.l., especially for the transect sites. Despite this rather low coverage for specific groups at high elevation, we concluded that the data were sufficiently complete to provide insight into diversity patterns along the elevation gradient.

For the abovementioned eight groups (all-, endemic-, native- and introduced species and four separate orders), alpha-diversity patterns were analysed by observed species richness, as well as total richness estimated by the Jacknife-1 estimator, and by mean richness per sample representing the first Hill number q0. Species richness per order was re-calculated over only indigenous (i.e., native and endemic) species. We tested these alpha-diversity patterns with linear regression models in R with elevation as the single predictor variable. Furthermore, three additional Hill numbers were investigated to assess evenness or dominance in community structure, namely q1—Shannon-Wiener (exp H’); q2—Simpson (1/D) and q3—Berger-Parker index (1/d) [63]. Gamma diversity was defined as the total diversity along the whole elevation gradient. All diversity indices were calculated with EstimateS software [61].

The sums of each species’ summer records per transect site were used to analyse beta-diversity patterns, using the R package BAT [64]. Total beta diversity and its replacement and richness difference partitions were derived from both incidence (presence/absence) and abundance-based pairwise comparisons, using the Jaccard dissimilarity index. We also tested Sorensen dissimilarity indexes which revealed highly similar patterns (not shown). Patterns over all pairs of sites and over adjacent pairs were both explored. Due to the low number of sites, only all pairs’ comparisons could be tested statistically with multiple regression on distance matrices (MRM) to account for spatial autocorrelation, using the R package ecodist [65,66]. The relative dominance of both beta-diversity components over all pairs was tested with Wilcoxon signed-rank tests.

## 3. Results

### 3.1. Altitudinal Ranges

A total of 134 species were recorded at the five transect sites; see Appendix A for a list of all species and their recorded altitudinal ranges. This included all the orders identified to species level, which represented almost 40% of the specimens collected in summer, but only about 25% of the specimens collected over the whole year and 9% of the specimens collected at 930 m a.s.l. (data from 2015, including Diptera, Lepidoptera and Hymenoptera but not Collembola and Acari). However, the orders identified to species level here represented 62% of the potential arthropod species pool on Terceira island and 53% of its endemic species [60]. Of these 134 species, 23 are endemic to the Azores archipelago, 53 are native non-endemic and 57 are introduced species, while one species has an uncertain status. The most species-rich orders were Coleoptera, Araneae, Hemiptera and Psocoptera with 44, 36, 28 and 13 species, respectively. A majority of species was found at a narrow elevational range of one or two sites.

Figure 2 shows the altitudinal distributions of all species and Araneae, Coleoptera, Hemiptera and Psocoptera along the elevation transect. Araneae showed an even distribution along the elevation gradient and a particularly high share of narrow-ranged species. For all species and for Coleoptera, Hemiptera and Psocoptera, lower elevations hosted the highest species diversity and most narrow-ranged species. This pattern was the strongest among introduced species, whereas endemic species occurred predominantly at higher elevations. Incidental records (singletons) were dominant among endemic species at low elevations (the first two sites) and represented five out of six introduced species at 930 m, but were uncommon among native and endemic species at mid-to-high elevations.

### 3.2. Alpha-Diversity Patterns

A linear species richness decrease was observed for all species on the five sites representing Terceira’s elevation gradient (linear model *p* = 0.013) (Figure 3a). This decrease was reflected in Coleoptera and Psocoptera species richness (linear model *p* = 0.0099 and 0.0054, respectively), but not in Hemiptera (linear model *p* = 0.078) and in Araneae species richness which peaked at 693 m a.s.l. (linear model *p* = 0.81) (see Figure 3b). Highly similar trends were observed for the four-season data across all 12 sampling sites. Coleoptera, Araneae at low elevations and Hemiptera at mid-low elevations showed the largest gains in species richness in this dataset compared to the transect data (Appendix A). Over all sites and over the elevation transect, Total (Jack1) species richness estimates revealed highly similar patterns of linear richness decrease (except for Araneae), supporting our judgments of sampling sufficiency (Appendix A). See Appendix A for all alpha-diversity measures and linear model coefficients.

Species richness of native non-endemic and introduced species along the elevation transect showed a linear decrease with elevation (linear model *p* = 0.026 resp. 0.0082), but endemic species richness was found to be highest at 693 m a.s.l. (linear model *p* = 0.45) (Figure 3c). These patterns were highly similar across four seasons for the 12 sampling sites and for total (Jack1) species richness estimates of both site selections (transect and all sites) (Appendix A). Alpha-diversity patterns of only indigenous species per order reflected that the high richness of native species at low elevation was to a large extent driven by Hemiptera, while in all orders but mainly in Coleoptera, the exclusion of introduced species reduced species richness mainly at low elevations (Appendix A). At 231 m a.s.l., species richness was slightly lower than expected by linear models for all, native and endemic species, as well as for indigenous Araneae, Coleoptera and Hemiptera, but was slightly higher for introduced species. Alpha-diversity measures and linear model coefficients are provided in Appendix A.

The mean species richness per sample (i.e., the first Hill number q0) showed high standard deviations but reflected the patterns of observed- and estimated-total species richness for all analysed groups (Appendix A). At 231 m a.s.l., mean species richness dropped for all, native and endemic species, but not for introduced species. This drop was seen in Hemiptera and not in Araneae, Coleoptera and Psocoptera, but it was apparent for indigenous species in all orders (Appendix A). Community evenness patterns, indicated by the second to fourth Hill numbers q2–q4, did not match the alpha-diversity patterns described above. For all species and native species, q2–q4 peaked at 404 m a.s.l. and reached minimum values at the low and high extremes of the elevation transect, while for introduced and endemic species, no clear elevational patterns were indicated. For Araneae, Coleoptera and Psocoptera, community evenness patterns roughly resembled species richness patterns, while for Hemiptera q2–q4 were very low across the transect (Appendix A).

### 3.3. Beta-Diversity Patterns

Beta diversity, defined as the difference between total (gamma) and mean (alpha) observed species diversity over the elevation transect, was the dominant component of total diversity for all species combined (64%) as well as Araneae, Coleoptera and Hemiptera and for native and introduced species. Alpha and beta diversity were almost equally important for Psocoptera and endemic species (Appendix A). Over adjacent sites, incidence-based beta diversity of all species showed only small fluctuations along the elevation transect (0.65–0.7). Higher beta-diversity values were found for Araneae at lower elevations and for Coleoptera at high elevation, while low beta-diversity values were found for Psocoptera (Appendix A, columns 1–4). Abundance-based beta diversity over adjacent sites was high for all species combined as well as each separate order (mean >0.85 for each group) but decreased with elevation for all species and Hemiptera (Appendix A, columns 1–4). Both incidence-based and abundance-based beta diversity over adjacent sites were high for introduced species, especially at higher elevations, and relatively low for endemic species, while for native species, both were very similar to all species. The replacement and richness difference components of both incidence-based and abundance-based beta diversity showed no clear trends along the elevation gradient but rather large fluctuations.

Incidence-based beta diversity increased with the distance in elevation between compared sites both for all species (Figure 4a) and for Coleoptera, Hemiptera and Psocoptera, while abundance-based beta diversity increased with the distance between sites only for Araneae and Coleoptera (Figure 4b and Appendix A). For all species and the different orders, replacement showed a weak response to the distance between sites, whereas incidence-based richness difference increased significantly with distance for all species as well as for Coleoptera and Psocoptera. Richness was the dominant component of beta diversity at large distances for all species and each order except Araneae. For both incidence-based and abundance-based beta diversity, replacement was dominant over richness for Araneae, and richness was dominant for Psocoptera over all distances, whereas both components were not notably different for all species, Coleoptera and Hemiptera (Appendix A).

For all species, native species and Hemiptera, markedly high abundance-based beta richness estimates but low replacement estimates were found for comparisons between the first site at 46 m a.s.l. and any other site (Figure 4b and Appendix A).

For native species, incidence-based total beta diversity and richness difference increased with distance and richness was the dominant component at large distances, while for endemic species, incidence-based total beta diversity and replacement increased with distance and replacement was the dominant component at large distances (Appendix A). For introduced species, abundance-based but not incidence-based total beta diversity increased with distance, while both incidence-based and abundance-based richness increased with distance and was the dominant component at large distances. Abundance-based replacement decreased with distance. Across all distances, richness was the dominant component of abundance-based beta diversity for both native and introduced species, while no component was significantly dominant in incidence-based beta diversity (Appendix A).

## 4. Discussion

### 4.1. Data Collection

As explained in detail in Borges et al. [47,49], Costa and Borges [58] and supported by sensitivity analyses in Matthews et al. [48], the SLAM sampling methodology is highly standardised and unlikely to cause notable differences in sampling intensity. Moreover, these traps were also very efficient in detecting rare endemic beetle species at low elevation exotic sites in Terceira island [67]. However, field circumstances, such as weather conditions, may have affected individual samples. Indeed, one sampling site where this affected the sampling persistently was excluded entirely, and a few other samples were excluded or missing because of damage to the traps by birds or wind. The remaining samples sustained an equal sampling of the selected transect sites over four summer seasons, except for the 930 m-site, for which only three samples were available. These three samples sustained high coverage of all species and most subgroups but a low coverage of Coleoptera and notably introduced species. Given that richness patterns were mostly determined on low-to-mid elevations, rarefaction procedures to compensate for this missing sample were not applied.

Over all seasons, at minimum 15 samples were available per elevation, but multiple sampling sites provided much higher numbers of samples at certain elevation bands, notably at 600–800 m a.s.l., requiring rarefaction of the data over all 12 sites. Sampling coverage estimates for this data were generally slightly higher than those for the transect summer records but were notably higher for introduced species, Coleoptera and Psocoptera at higher elevations, as well as for Araneae (Appendix A). Despite these differences in coverage, highly similar patterns were found for the two datasets (transect summer records and all-season records over all sites), as well as for observed- and estimated-total richness estimates for both datasets. This supports our judgment that the different groups were sampled sufficiently to support across-gradient and across-group conclusions, although incidental low coverage estimates ask for caution when making statements based on single sites.

The selected transect on the western slope of Terceira represents the island’s only part where native vegetation still persists at lower elevations, albeit in a more disturbed state than at high elevation [28]. The 693 m-site is situated several kilometers to the east but is still situated on the Santa Bárbara volcanic complex in the west of the island (Figure 1). The other sites at this elevation are all located further towards the east and are included to test if species richness patterns persist on a larger spatial scale, representing the total area of native vegetation on Terceira island.

The data presented in this study included the species from 13 arthropod orders, but not the abundant and species-rich orders Diptera, Hymenoptera and Lepidoptera. The included orders represent a minority of the total SLAM trap catch that was almost 40% in summer, but 25% over the whole year, and decreased with elevation to only 9% at 930 m a.s.l. Diptera, Hymenoptera and Lepidoptera were thus relatively more abundant at higher elevations and might also show different diversity patterns than those presented here. The included orders do, however, represent the majority of arthropod species known from Terceira island [60]. It is therefore probable, but unconfirmed, that the elevation patterns presented here also apply to Terceira’s total arthropod diversity.

### 4.2. Elevational Diversity Patterns

Observed- and estimated-total richness patterns on the elevation transect revealed a monotonic decrease in species richness with elevation for all species and for Coleoptera and Psocoptera, and indicated a similar pattern for Hemiptera, but showed no clear pattern for Araneae. The consistent, highly similar patterns over all 12 sites showed that this was a general pattern on Terceira island. In contrast to our expectations, no clear effects of disturbance and the transition from disturbed to pristine vegetation at mid-elevation on total species richness were indicated. Instead, monotonic decreases are mainly temperature-driven. They may be especially common in temperate regions and ectotherm groups where a stronger temperature effect can be expected and on wet mountains that lack strong moisture constraints at lower elevations [6,17,24] (however, see [27]). Next to temperature and precipitation, radiation input, oxygen availability and wind turbulence influence insect occurrence on higher elevations [15]. It is therefore likely that deteriorating climatic circumstances, i.e., harsh climatic conditions (sensu [68]), are the main determinant of the decreasing arthropod species richness on the humid and temperate Terceira elevation gradient (see also [46]).

Elevational richness patterns may, however, differ between orders, as was shown by Araneae in this study, and even between different clades within orders, depending, e.g., on differing diets or dispersal strategies. Further investigation of such patterns could reveal further insight into the ecological drivers of species’ ranges and diversity patterns along elevation gradients but was outside the scope of this study.

We found contrasting patterns among endemic species richness, which peaked at higher elevations and introduced species richness which strongly decreased with elevation, corresponding to our expectations and previous studies, e.g., [1,20,69,70]. However, contrary to our expectations, native non-endemic species richness also decreased with elevation in close resemblance of all species combined. The high endemic species richness at high elevation could be driven by both increasing isolation, which induces speciation and endemism [1,69,71], or by the high human disturbance and occurrence of introduced species at lower elevations [42,72]. Elevation strongly coincided with human disturbance and alien plant occurrence on Terceira island, and near-pristine native forests are confined to elevations above 500 m [42,46]. Contrastingly, introduced species can profit from both decreased isolation, increased human disturbance, low competition with endemic species and alien plant occurrence at low elevations, which likely drive the strong low-elevation preference of introduced species [48,49,70] (however, see [72]). Nevertheless, a few introduced species were recorded only at high elevations. These species originate from cool temperate climates (see also [29]) and indicate that introduced species are also able to reach the isolated natural forests at high elevations. These species are most common in other Azorean habitats and behaving as tourists in native forests in a dynamic source–sink process [48,50]. Native non-endemic species richness seemed less affected by human disturbance and alien plant occurrence than by climatic circumstances on the elevation gradient. Species richness of indigenous (native and endemic) species per order differed most strongly from the total species richness per order at low elevations, especially for Coleoptera (with many introduced species) and less so for Hemiptera (with few introduced species). However, like all species and native non-endemic species richness, indigenous species richness still decreased with elevation in each order except for spiders (Araneae) and is thus probably mainly temperature-driven.

The observed drop in mean richness per sample, i.e., the first Hill number q1, at 231 m a.s.l., is likely reflecting the influence of human disturbance and alien plant growth at this site, which was the most disturbed along the transect. Indeed, this drop was present among native and endemic species but not among introduced species and was reflected in each order only among indigenous species. The second to fourth Hill numbers contrasted to other species richness patterns for all and native species as they peaked at mid-elevation instead of low elevation. This suggests that community structures might be less balanced at both extremes of the elevation gradient with the presence of more rare species and some levels of species dominance. However, Hill number estimates were strongly influenced by the extreme abundance of single dominant species, especially of the bug *Plinthisus minutissimus* (Fieber) (Hemiptera) at 46 m a.s.l. with over 5000 individuals caught, i.e., over 70% of all specimens at this elevation and over 50% of all specimens caught at the entire elevation transect in summer. A few Hemiptera species were the most abundant species across the transect, which was reflected by the very low community evenness of this order and by the fact that for Araneae, Coleoptera and Psocoptera Hill numbers did not drop at low elevation. We thus concluded that the divergent Hill number patterns were mainly an artefact of this extreme abundance of single species.

### 4.3. Beta-Diversity Analyses

The general dominance of beta diversity over alpha diversity and the high rates of both incidence-based and abundance-based beta diversity along the elevation gradient revealed the importance of differences between sites in shaping elevational patterns of arthropod diversity on Terceira island. Comparisons between alpha and beta diversity should be made with caution as beta-diversity estimates may be enhanced by incomplete sampling [73], but beta diversity is clearly a major driver of total diversity on the Terceira elevation gradient. Abundance-based beta diversity was higher than incidence-based beta diversity for all groups, indicating that different elevations differed even more in species abundance than in diversity, which may reflect the decrease in productivity with elevation [1]. However, abundance-based estimates appeared to be strongly influenced by the high abundance of single species, as described above, especially among all and native species and Hemiptera. Therefore, in this study, incidence-based estimates provided a more robust basis to compare beta-diversity differences between groups. As we predicted, the highest beta-diversity rates were found for introduced species where source–sink dynamic play an important role [48,49]. Beta diversity was also high for other groups with many narrow-ranged species, i.e., Araneae and Coleoptera, but was relatively low for endemic species and for Hemiptera and Psocoptera that include few narrow-ranged species (Figure 2).

The relatively short length of our elevation transect limited our ability to analyse elevational trends in beta diversity among adjacent sites (i.e., four comparisons), as has been performed on longer transects, e.g., [38,39]. Nevertheless, we measured that total beta diversity was rather stable along the elevation transect and showed no signs of ecotone effects at, e.g., the transition from disturbed to more pristine vegetation at 400–600 m a.s.l. We found a high variability of the replacement and richness components of beta diversity along the elevation transect, which corresponds to other studies, e.g., [38,39] and may reflect differing driving factors across taxa. The significant increase in total beta diversity with elevational distance for all species and across all groups, except Araneae and introduced species, corresponds to the theory of “distance decay of similarity between sites” [74]. For Araneae and introduced species, as well as for most abundance-based beta diversity estimates, total beta diversity tended to increase with distance as well but was already close to 1 at low distances (Appendix A). This increase in beta diversity with distance was driven by increasing richness differences for all species and most groups, except Araneae and endemic species, reflecting the heterogeneity of the habitat matrix surrounding the sampled plots.

We expected replacement to be the dominant component of beta diversity, but we only found a dominance of replacement for Araneae, reflecting the large share of narrow-ranged species and the weak response of species richness to elevation in this group. No component was dominant across all species, but richness difference was dominant for Psocoptera, reflecting the nested pattern of decreasing richness with elevation in this group (Figure 2), as well as for abundance-based estimates of native and introduced species. Beta partitioning was, however, dependent on elevational distance decay which induces contrasting responses of species replacement and richness difference. Richness differences increased with elevational distance as species diversity decreased with elevation in all groups except Araneae and endemic species, but species replacement remained stable or decreased with elevational distance. Richness rates were indeed higher than replacement rates over large distances in all groups except Araneae and endemic species, whereas, over adjacent sites, species replacement was higher than richness difference for all species, Araneae and Coleoptera and introduced species. These groups had the highest total beta diversity and showed considerable species replacement among narrow-ranged species (Figure 2). The importance of species replacement in beta diversity as noted by previous studies [1,39] was most clear in these groups, whereas monotonous decreases in species richness with elevation induced large richness differences between sites across all species and all groups except Araneae and endemic species. Both components are thus important on the Terceira elevation gradient.

## 5. Conclusions

Species richness of all analysed arthropod species and Coleoptera, Hemiptera and Psocoptera on Terceira island, Azores (Portugal) showed a monotonic decrease with elevation. This pattern is likely driven by climatic conditions, as is common among ectotherm groups and in humid and temperate systems worldwide. Native non-endemic species closely follow this pattern. By contrast, species richness of Araneae and endemic species was highest at higher elevations, leading to increased endemism at high elevation where both more isolated communities and more pristine natural habitats exist. Introduced species showed the strongest decrease in species richness with elevation, which reflects the decreasing disturbance with elevation at our research location. The low mean richness per sample across all groups except introduced species at 231 m a.s.l., corresponding to the most disturbed site along the elevation gradient, also indicates the influence of matrix disturbance and alien plant growth on arthropod species richness on the Terceira elevation gradient.

Beta diversity, i.e., the community difference between elevations, was the major component of the total diversity in all groups except Psocoptera and endemic species. It was not related to elevation but increased with the distance between sites (i.e., distance decay). Beta diversity was the highest for introduced species driven by source–sink dynamics, as well as for Araneae and Coleoptera, which include many narrow-ranged species. Both replacement and richness differences were important drivers of total beta diversity. Replacement was dominant over adjacent sites for all species and the groups with the highest beta diversity (Araneae, Coleoptera and introduced species). Richness differences increased with distance and were dominant over large distances in all groups except Araneae and endemic species, reflecting the decreases in species richness with elevation.

## Figures and Tables

**Figure 1 insects-12-00936-f001:**
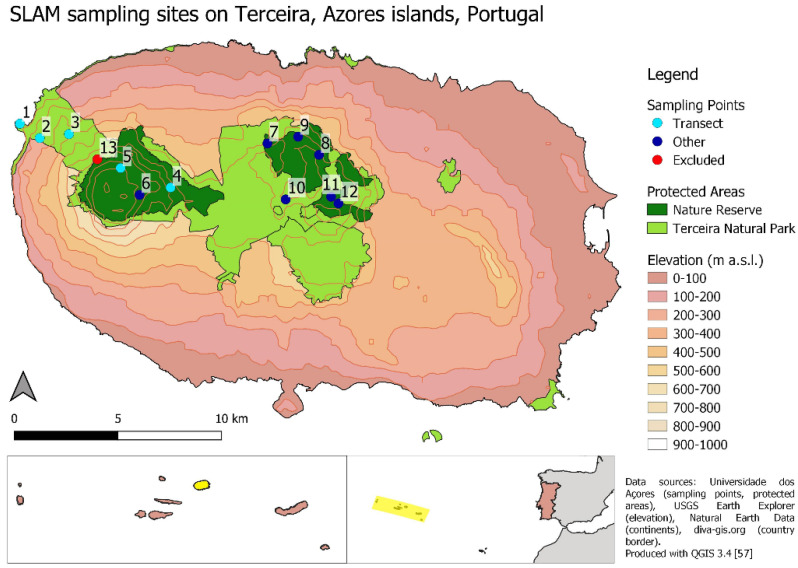
Map of Terceira island, showing the island’s elevation above sea level and the location of protected areas that represent the remaining distribution of semi-natural vegetation (Terceira Natural Park) and native laurel forests (Nature Reserve) [57]. The 12 SLAM sampling sites analysed in this study are indicated in light blue (transect sites representing the elevation gradient at 46 m (1), 231 m (2), 404 m (3), 693 m (4) and 930 m (5) a.s.l.) and dark blue (other sites). One site was excluded due to persisting field problems (red). The two insets at the bottom indicate the location of Terceira island within the Azores archipelago (**left**) and the location of the Azores in the Atlantic ocean west of Portugal (**right**).

**Figure 2 insects-12-00936-f002:**
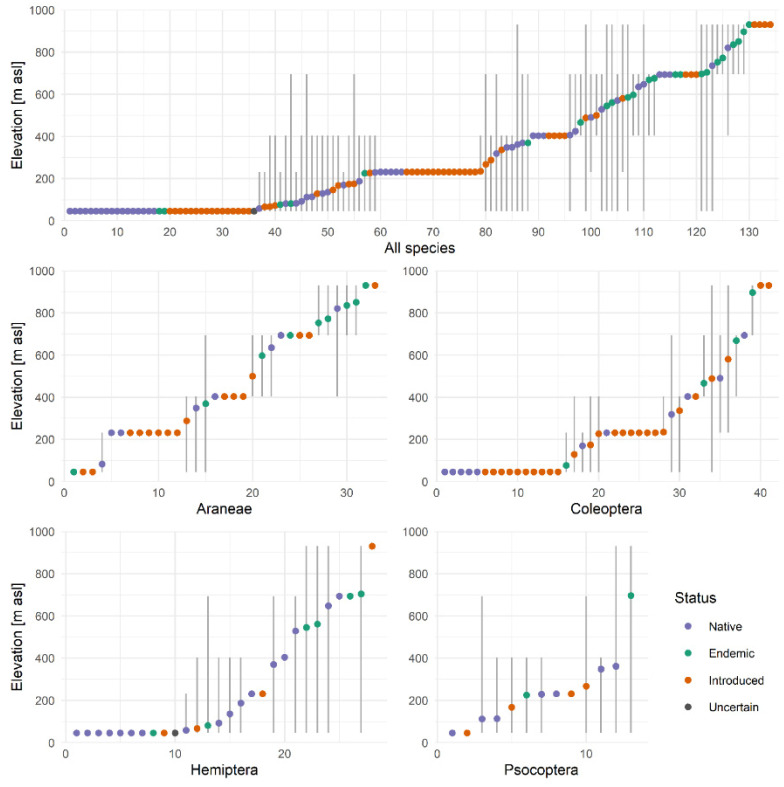
Altitudinal ranges along the elevation gradient of Terceira island, represented by five sampling sites at 46, 231, 404, 693 and 930 m a.s.l., for all species and for the orders Araneae, Coleoptera, Hemiptera and Psocoptera (species numbers are on the *x*-axes). The graphs show each species’ range, assuming continuous ranges between the minimum and maximum recorded elevations, and the abundance-based mean elevation of all records. Species are sorted by mean elevation. Colours indicate native (but non-endemic) species (blue), endemic species (green) and introduced species (orange), while one Hemiptera species has an uncertain status (grey). Many species were only recorded at one site. The highest species richness among different orders was found at low elevations, especially for introduced species, but not for endemic species and Araneae. Appendix A lists all species shown here.

**Figure 3 insects-12-00936-f003:**
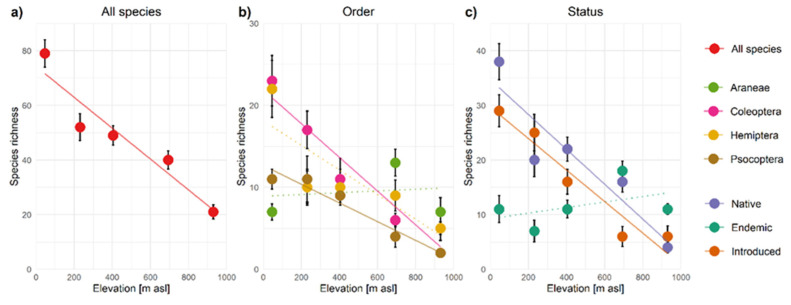
Observed species richness (S) patterns over the five sites on the elevation transect for: (**a**) all species, (**b**) four different orders and (**c**) native, endemic and introduced species. The error bars indicate standard deviations across four samples per elevation. The linear trendlines are shown as solid lines for species groups where *p*-values are <0.05 and indicated with dotted lines when *p* ≥ 0.05.

**Figure 4 insects-12-00936-f004:**
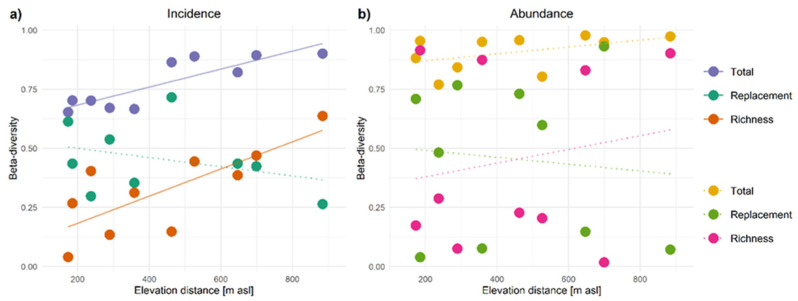
Patterns of total beta diversity and its replacement and richness components over the elevational distance (i.e., difference in m a.s.l.) between sites on the five-site elevation transect for all species, providing; (**a**) incidence-based, and (**b**) abundance-based beta diversity estimates. The linear trendlines are shown as solid lines for species groups where *p*-values are <0.05 and indicated with dotted lines when *p* ≥ 0.05.

## Data Availability

The data presented in this study and R scripts presenting the data processing and diversity analyses are publicly available via Zenodo: DOI 10.5281/zenodo.5567768 (https://zenodo.org/record/5567768#.YWfVWhrP2F5 accessed on 14 October 2021).

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
