# Peer review of "A Small-Scale Analysis of Elevational Species Richness and Beta Diversity Patterns of Arthropods on an Oceanic Island (Terceira, Azores)"

_insects, 2021, doi:10.3390/insects12100936_

Round 1

Reviewer 1 Report

Dear authors

You did not respond to my main remarks, or rather replied that you will take my remarks into account in a future article. Nevertheless, this version of this manuscript is much better than the initial one and therefore, as I don't feel necessary to block artificially this work, I think this article can be published in its present form.

Reviewer 2 Report

The authors have addressed all points I had raised in my review in a very satisfactory manner. The revised version also presents more context about the island and study sites, as well as on the trap type. To my opinion this is a very well performed revision that does not leave any queries open from my perspective.

Reviewer 3 Report

Authors improved the ms since the first version. Therefore, it can be accepted for publication on Insects.

This manuscript is a resubmission of an earlier submission. The following is a list of the peer review reports and author responses from that submission.

Round 1

Reviewer 1 Report

This paper presents yet another case study on biodiversity patterns along a single, un-replicated elevational transect on an oceanic island, in the archipelago of the Azores. Arthropods were sampled using Malaise traps, in forested sites, at elevational distances of approx. 200 m between sampling sites. The design of the study is okay, with regard to the limitations set by the extent of remaining forest and the rather small elevational gradient that exists on the studied island. Traps were run over a series of years, with trap catch time intervals of roughly 3 months duration.

The study is placed in the context of the vast literature that has been assembled on biodiversity patterns along elevational gradients. Overall the study is well written, and much literature has been consulted, even though I did not always get the feeling that all of these references were the most relevant. However, this is a minor point.

Apart from a couple of specific suggestions, I recognized two major issues that require to be addressed when revising the paper.

First, the authors only evaluated and analysed collected specimens from 4 taxa. The selection of these taxa really sounds arbitrary: beetles, spiders, psocopterans and hemipterans. Thus, this is a wild mixture of herbivores, predators and detritivores. However, nowhere in the paper did I see how large, or small, this fraction of evaluated taxa was, in relation to total trap catches. Did these 4 focal groups account for 10, 50, or 90 % of total catch? Did this representation vary across the elevational gradient? It is particularly disturbing, for a study based on Malaise traps (which target flying insects) that apart from Coleoptera, all really large orders of pterygote insects were neglected (apparently because of the lack of taxonomic expertise in these groups).

Second, various analyses presented in the paper rest on estimates of “sample completeness”. These estimates are based on extrapolating asymptotic species numbers to “infinite” samples. However, this concept has become a bit outdated by now. I very strongly suggest to replace all these analyses by estimates of sample coverage, as they are for example available in the iNext statistical package (https://cran.r-project.org/web/packages/iNEXT/index.html). Sample coverage has many advantages over putative completeness, especially since the estimates of “total” richness all have rather large levels of uncertainty, which need to be accounted for.

Finally, there is a general tendency of the authors to exaggerate about their study. For example, the write about “high diversity”, but the sampled island fauna clearly is NOT at all “highly diverse”. Or they write about “total arthropods”, but in fact they omitted large fractions of arthropod diversity. In all these places, a careful re-wording is needed.

Specific comments, in the sequence as they popped up when reading the paper.

L 25: later on one learns that traps were only set up to 930 m elevation, please provide consistent information throughout the paper!

L 45: alternatively, gamma diversity may be seen as the product of alpha and beta diversity. Why did you decide for your additive concept?

L 48: see the work by Tuomisto, published in Ecography about 10 years ago: there are various meanings available of 'beta diversity' on the market, please tell precisely whether you are talking about differentiation or proportional 'beta-div' or something else.

Tuomisto, H. (2010). A diversity of beta diversities: straightening up a concept gone awry. Part 1. Defining beta diversity as a function of alpha and gamma diversity. Ecography, 33(1), 2-22.

L 58: I disagree - the same applies to mountain ranges in the equatorial tropics, e.g. the Andes in S America, or Mt Kinabalu in Borneo, etc. In other words: your emphasis on temperate regions is just not correct here.

L 119: The wording is unclear here. Were traps placed out every year? Please tell precisely. Only later in the methods section one gets a better feeling of what you really did.

L 185: This is a case in point of my above criticism. Your wording suggests to the quick reader that you have analysed “all arthropods”, but in reality your paper deals only with the few taxa that could be taxonomically identified. Please use precise and honest wording.

L 198: see above comment, better use sample coverage rather than putative "completeness", since any of the extrapolation estimates, however “good” they might be, have (often: substantial) confidence limits, which otherwise would need to be incorporated into “completeness” estimates (usually derived as the quotient of observed vs. “total” richness).

L 210: an overall richness of 134 species, across all these selected arthropod orders, does NOT indicate a highly "diverse" fauna! Rather, as one would expect for an isolated small oceanic island, this points to a really low diversity of the local fauna.

L 233: how did you deal with the stochastic non-independence of all these pairwise similarity comparisons? The stochastic independence of all data points, which is violated by your data structure, is an essential prerequisite of any standard (e.g. OLS) regression analyses. Hence, rather some mode of analysis that takes into account spatial autocorrelation seems required here.

L 238: see my remark above: this is NOT a truly "diverse" fauna, and accordingly your wording should by toned down in the introduction and elsewhere. For example, 44 beetle species sampled over multiple years are NOT a rich fauna, by all standards!

L 361: the reader should learn about the fraction of SLAM samples that you cover with your here presented analysis of just 4 arthropod taxa, especially since many orders of flying insects were NOT analysed. Did the four target taxa represent 10, 50, or 90 % of total SLAM catches per site (on average)? Did that representation vary across the elevational gradient?

L 503: that is again a misleading wording. Your analysis does not tell anything about "all arthropods", but at best about a composite value for 4 arbitrarily selected arthropod taxa, while ignoring multiple species-rich holometabolous insect orders, likely in excess of all what you have analysed. Please use a more appropriate, precise and adequate wording, here and throughout the paper.

Author Response

REFEREE 1

This paper presents yet another case study on biodiversity patterns along a single, un-replicated elevational transect on an oceanic island, in the archipelago of the Azores. Arthropods were sampled using Malaise traps, in forested sites, at elevational distances of approx. 200 m between sampling sites. The design of the study is okay, with regard to the limitations set by the extent of remaining forest and the rather small elevational gradient that exists on the studied island. Traps were run over a series of years, with trap catch time intervals of roughly 3 months duration.

The study is placed in the context of the vast literature that has been assembled on biodiversity patterns along elevational gradients. Overall the study is well written, and much literature has been consulted, even though I did not always get the feeling that all of these references were the most relevant. However, this is a minor point.

RESPONSE: Thank you for this reflection. We agree with your view and acknowledge the limitations of our study design that you have mentioned here. We thank you very much for your careful critical evaluation of our manuscript and we have carefully considered your critics.

Apart from a couple of specific suggestions, I recognized two major issues that require to be addressed when revising the paper.

First, the authors only evaluated and analysed collected specimens from 4 taxa. The selection of these taxa really sounds arbitrary: beetles, spiders, psocopterans and hemipterans. Thus, this is a wild mixture of herbivores, predators and detritivores. However, nowhere in the paper did I see how large, or small, this fraction of evaluated taxa was, in relation to total trap catches. Did these 4 focal groups account for 10, 50, or 90 % of total catch? Did this representation vary across the elevational gradient? It is particularly disturbing, for a study based on Malaise traps (which target flying insects) that apart from Coleoptera, all really large orders of pterygote insects were neglected (apparently because of the lack of taxonomic expertise in these groups).

RESPONSE: Thank you very much for this excellent suggestion. We have evaluated and analysed all orders amenable for a rapid morphospecies identification, which does not include the three hyper-diverse orders Diptera, Hymenoptera and Lepidoptera (mostly moths) due to taxonomic difficulties. However only the orders Araneae, Coleoptera, Hemiptera and Psocoptera were diverse enough to allow analyses of order-specific diversity patterns. The abundance of these groups is due to the fact that in this habitat (native Azorean forest), humid conditions and extensive epiphytic growth on the tree branches where the traps stand on result in a relatively high catch ratio of terrestrial and arboreal species relative to flying species. We have clarified this species selection in the Materials & Methods section (2.2 and 2.3).
We value your suggestion to show the fraction of the evaluated taxa relative to the total catches, which are now provided in the Results section (3.1).  We could only assess this over the year 2015 when specimen counts of Diptera, Lepidoptera and Hymenoptera (but not Collembola and Acari) were available, but we could assess the variation across the elevation transect. Moreover, we assessed the ratio of the included orders relative to the total arthropod species pool present on Terceira island, which was published by Borges et al. in 2010 and kept up to date. The evaluated orders represent a minority of specimens in the trap catches but a majority of arthropod species present on Terceira island. The representation is now discussed more elaborately in the manuscript. However, there is no evidence that any of the non-selected orders will show notably different patterns than presented here.

Second, various analyses presented in the paper rest on estimates of “sample completeness”. These estimates are based on extrapolating asymptotic species numbers to “infinite” samples. However, this concept has become a bit outdated by now. I very strongly suggest to replace all these analyses by estimates of sample coverage, as they are for example available in the iNext statistical package (https://cran.r-project.org/web/packages/iNEXT/index.html). Sample coverage has many advantages over putative completeness, especially since the estimates of “total” richness all have rather large levels of uncertainty, which need to be accounted for.

RESPONSE: Thank you for this important point. We have revised our assessment of sampling completeness and now use coverage estimates instead of estimates of the extrapolated total species richness. We generally obtained high coverage levels across the gradient for the different groups, but a rather low coverage at 930 m a.s.l. for Coleoptera and introduced species. We discuss these results in 4.1.

Finally, there is a general tendency of the authors to exaggerate about their study. For example, the write about “high diversity”, but the sampled island fauna clearly is NOT at all “highly diverse”. Or they write about “total arthropods”, but in fact they omitted large fractions of arthropod diversity. In all these places, a careful re-wording is needed.

RESPONSE: Thank you for this remark. We fully agree with your view that e.g. the Azorean arthropod fauna is not at all ‘highly diverse’ and we don’t want to suggest that in this manuscript.  We re-wrote these phrases throughout the manuscript.

Specific comments, in the sequence as they popped up when reading the paper.

L 25: later on one learns that traps were only set up to 930 m elevation, please provide consistent information throughout the paper!

RESPONSE: We have changed this sentence to make clear that 0-1021 m a.s.l. represents the whole elevation gradient on Terceira island, which should indeed be clear from the start. We used five traps (46 – 930 m. asl.) to represent this gradient. We avoid the high and low extremes to avoid adverse conditions here (e.g. exposure to strong wind and waves/salt spray).  

L 45: alternatively, gamma diversity may be seen as the product of alpha and beta diversity. Why did you decide for your additive concept?

RESPONSE: We have explained this decision in the manuscript. MacArthur et al. (1966) suggested an additive partition of diversity instead of the multiplicative partition proposed by Whittaker, and such a partition received much more attention when Lande (1996) linked the concept with the alpha, beta and gamma terms of Whittaker, proposing that gamma = alpha + beta. This meant that beta diversity could be measured in terms of species numbers in the same way as alpha and gamma, instead of being the unitless measure of turnover resulting from multiplicative partition. Since then, several authors have shown the utility of using an additive instead of a multiplicative partition for some analyses (see for a review  Veech et al. 2002. The additive partitioning of species diversity: recent revival of an old idea. Oikos, 99(1), 3-9).

L 48: see the work by Tuomisto, published in Ecography about 10 years ago: there are various meanings available of 'beta diversity' on the market, please tell precisely whether you are talking about differentiation or proportional 'beta-div' or something else.

Tuomisto, H. (2010). A diversity of beta diversities: straightening up a concept gone awry. Part 1. Defining beta diversity as a function of alpha and gamma diversity. Ecography, 33(1), 2-22.

RESPONSE: We have clarified which definitions of alpha, beta and gamma diversity are used in this manuscript (see also response above). 

L 58: I disagree - the same applies to mountain ranges in the equatorial tropics, e.g. the Andes in S America, or Mt Kinabalu in Borneo, etc. In other words: your emphasis on temperate regions is just not correct here.

RESPONSE: Thank you for this remark in which you are right. We meant to state not that elevation-driven speciation is especially apparent in temperate regions, but that elevation may relatively speaking be an especially important driver of speciation in temperate regions, as other driving factors are often lacking. We have adapted this sentence.

L 119: The wording is unclear here. Were traps placed out every year? Please tell precisely. Only later in the methods section one gets a better feeling of what you really did.

RESPONSE: We have clarified this part. 

L 185: This is a case in point of my above criticism. Your wording suggests to the quick reader that you have analysed “all arthropods”, but in reality your paper deals only with the few taxa that could be taxonomically identified. Please use precise and honest wording.

RESPONSE: Thank you for pointing this out. We don’t want the manuscript to contain incomplete or misleading phrases, so we have specified which orders specifically are excluded and included under 2.1 (excluded) and 2.2 (included).  

L 198: see above comment, better use sample coverage rather than putative "completeness", since any of the extrapolation estimates, however “good” they might be, have (often: substantial) confidence limits, which otherwise would need to be incorporated into “completeness” estimates (usually derived as the quotient of observed vs. “total” richness).

RESPONSE: See above 

L 210: an overall richness of 134 species, across all these selected arthropod orders, does NOT indicate a highly "diverse" fauna! Rather, as one would expect for an isolated small oceanic island, this points to a really low diversity of the local fauna.

RESPONSE: See above: we have avoided the ‘highly diverse’ label.  

L 233: how did you deal with the stochastic non-independence of all these pairwise similarity comparisons? The stochastic independence of all data points, which is violated by your data structure, is an essential prerequisite of any standard (e.g. OLS) regression analyses. Hence, rather some mode of analysis that takes into account spatial autocorrelation seems required here.

RESPONSE: Thank you for pointing this out, you are right that the datapoints analyzed here are not independent. We now used multiple regression on distance matrices (MRM) to test the beta-diversity patterns while taking into account spatial autocorrelation.

L 238: see my remark above: this is NOT a truly "diverse" fauna, and accordingly your wording should by toned down in the introduction and elsewhere. For example, 44 beetle species sampled over multiple years are NOT a rich fauna, by all standards!

RESPONSE: See above: we now avoid the label ‘diverse’ throughout the manuscript.

L 361: the reader should learn about the fraction of SLAM samples that you cover with your here presented analysis of just 4 arthropod taxa, especially since many orders of flying insects were NOT analysed. Did the four target taxa represent 10, 50, or 90 % of total SLAM catches per site (on average)? Did that representation vary across the elevational gradient?

RESPONSE: See above: Thank you for this excellent suggestion. We have added the ratio of species that the four target taxa represent per site under 3.1. Generally, these taxa represent the majority of specimens collected and identified to species level. Moreover, two of the selected taxa (beetles, spiders) have almost 40% of the Azorean arthropod endemics and therefore are good candidates for investigating macroecological patterns in Azores as representatives of the native fauna. Moreover, as now we mention in the discussion these traps were also very efficient in detecting rare endemic beetle species at low elevation exotic sites in Terceira island  [Tsafack, N., Fattorini, S., Boieiro, M., Rigal, F., Ros-Prieto, A., Ferreira, M.T. & Borges, P.A.V. (2021). The role of small lowland patches of exotic forests as refuges of rare endemic azorean arthropods. Diversity, 13(9): 443. DOI:10.3390/d13090443] 

L 503: that is again a misleading wording. Your analysis does not tell anything about "all arthropods", but at best about a composite value for 4 arbitrarily selected arthropod taxa, while ignoring multiple species-rich holometabolous insect orders, likely in excess of all what you have analysed. Please use a more appropriate, precise and adequate wording, here and throughout the paper.

RESPONSE: We have changed the terms ‘all arthropods’ to ‘all analysed species’ here and throughout the paper. This species selection does not just represent the four target orders together but also includes several smaller orders. This was not clear from the manuscript but is now explained under 2.2.

Reviewer 2 Report

Review’s comments Manuscript Number: Insects-1337768 General comments The manuscript entitled “A small-scale analysis of elevational species richness and beta diversity patterns of ~~~~~~” dealt with arthropods biodiversity in oceanic island. The manuscript focused on analysis of diversity patterns and offered only few information on species name, species composition of forest and sampling methods. 1. What is difference with SLAM trap and Malaise trap? And you mentioned that sampling intervals were roughly 90 days. Many insect samples can be decomposed without anti-decomposition chemicals. Besides, net of spider in the trap affects sampling efficiency of the trap. It is necessary to include sampling methods in details. 2. Why you selected Araneae, Coleoptera, Hemiptera and Psocoptera? As you know, many kinds of insects were collected by Malaise trap. I wonder whether some insects belonged to Hemiptera such as aphids were collected by trap and how you identified. These are easily decomposed in summer season. 3. More information of plant species composition in your site is necessary. As you know, diversity of insects was affected by plant diversity. To properly understand your survey, it is necessary to include information on plant (trees and understory vegetation) species composition. 4. Species information of spiders and insects is also necessary. Changes of species in according to elevation should be described and discussed. 5. More statistical information is added. For example, only P values were presented in the manuscript. F values, degree of freedoms, intercepts and slopes are added in the manuscript. Please see the pdf file attached to find minor corrections.

Author Response

REFERRE 2

Review’s comments Manuscript Number: Insects-1337768 General comments The manuscript entitled “A small-scale analysis of elevational species richness and beta diversity patterns of ~~~~~~” dealt with arthropods biodiversity in oceanic island. The manuscript focused on analysis of diversity patterns and offered only few information on species name, species composition of forest and sampling methods. 1. What is difference with SLAM trap and Malaise trap? And you mentioned that sampling intervals were roughly 90 days. Many insect samples can be decomposed without anti-decomposition chemicals. Besides, net of spider in the trap affects sampling efficiency of the trap. It is necessary to include sampling methods in details. 2. Why you selected Araneae, Coleoptera, Hemiptera and Psocoptera? As you know, many kinds of insects were collected by Malaise trap. I wonder whether some insects belonged to Hemiptera such as aphids were collected by trap and how you identified. These are easily decomposed in summer season. 3. More information of plant species composition in your site is necessary. As you know, diversity of insects was affected by plant diversity. To properly understand your survey, it is necessary to include information on plant (trees and understory vegetation) species composition. 4. Species information of spiders and insects is also necessary. Changes of species in according to elevation should be described and discussed. 5. More statistical information is added. For example, only P values were presented in the manuscript. F values, degree of freedoms, intercepts and slopes are added in the manuscript. Please see the pdf file attached to find minor corrections.

RESPONSE: Thank you very much for pointing out several points where important information is lacking. We have provided more information on the trapping method (specifically SLAM vs Malaise), and explain how decomposition is handled (SLAM trap recipient was filled with propylene glycol (pure 1,2-PROPANODIOL) to kill and conserve the samples between periodic collections, safeguarding DNA preservation for genetic analysis also). We have furthermore explained in more detail why these orders are selected (i.e. these are the most species-rich orders that provided enough diversity for independent analyses) and what percentage of the total number of specimens they represent. Aphids were identified based on our reference collection, and when not possible identified as morphospecies. We explain in short the general plant species composition of the Azorean native forest and differences along the transect. We have included in the supplementary materials a list of the included arthropod species with information on status and elevational range, but our data does not allow in-depth assessment of the distribution of individual species. Additional statistical information is provided in the supplementary information. We have now included the F-statistic in these tables and we refer more clearly to this in the manuscript. 

Reviewer 3 Report

Review Reinier de Vries 2021 Insects

Hereby my comments, This article is nice but lacks important points in order to properly interpret the results.

Line 84: A point already discussed in the literature is that beta-diversity is also governed by the rate of species dispersal. I think Cardoso et al. talk about such a factor in their articles with spiders and alternatively, you can also use this article which compares two groups of arthropods with very different mobilities:

 Luque C, Legal L, Winterton P, Mariano N.A, Gers C (2011) Illustration of the Structure of Arthropod Assemblages (Collembola and Lepidoptera) in Different Forest Types: An Example in the French Pyrenees. Diversity-Basel.

Line 109: Can you provide some main characteristics of this native Azorean forest (dominant trees, main fauna).

Line 160: Why exclude them? I can understand for Diptera which are sometimes difficult to identify but Madeira Lepidoptera are all well known (or do you consider adults too mobile?) And I guess very few Amphipods and Isopods species are presented.

Line 172: No control traps in perturbed areas?

Line 187: Maybe list the species here? If there is not too much this information can be nice to better illustrate your sampling.

Line 244: All Aranea are predators, while only a part of Coleoptera and Hemiptera, Psocoptera are not. Do you discuss this point later? This may partly explain your distribution results.

Figure 2 please use a more distinct color to separate native from uncertain even if only one dot is present.

Figure 3 and figure 4: A little confused for me. A comparison by order may be missing (native Aranea + endemic versus introduced and the same for the other orders). Finally, a main point is to check whether native species (including endemic species) have a different distribution pattern from that introduced. The same goes for comparisons of incidence and abundance. Don't you think that by combining order + feeding type (i.e. predatory or not) + introduced / native (including endemic) you might not achieve a better resolution (I might be completely wrong, but can you check it)?

Beta-diversity results. Same thing… I think you have to compare the values ​​for native Aranea versus introduced Aranea and so on (for each order) to be able to compare your results. I'm not convinced that an overall comparison of introduced / native / endemic arthropods is very informative as your different guilds may have very different dispersal abilities depending on the diet.

Line 407 to 429. I broadly agree with your conclusions but as mentioned before you cannot conclude with precision without taking into account the rate of dispersal as a function of the order (and even among orders: non-flying beetles will not disperse. as much as the flying ones) and the guild type. Most of the time, predators have a larger territory than phytophagous. Perhaps to provide detailed examples of some selected species such as your dominant seedeater: Plinthisus minutissimus, which is a very wide range species.

Line 482 to 487: Again, these differences could be due to the dispersal abilities of the species. Low dispersal species tend to be governed by replacement of species, while mobile species are more dependent on changes in species diversity.

Author Response

REFEREE 3

Hereby my comments, This article is nice but lacks important points in order to properly interpret the results.

Line 84: A point already discussed in the literature is that beta-diversity is also governed by the rate of species dispersal. I think Cardoso et al. talk about such a factor in their articles with spiders and alternatively, you can also use this article which compares two groups of arthropods with very different mobilities:

 Luque C, Legal L, Winterton P, Mariano N.A, Gers C (2011) Illustration of the Structure of Arthropod Assemblages (Collembola and Lepidoptera) in Different Forest Types: An Example in the French Pyrenees. Diversity-Basel.

RESPONSE: Thanks for pointing this out. We have mentioned this topic in the discussion.

Line 109: Can you provide some main characteristics of this native Azorean forest (dominant trees, main fauna).

RESPONSE: We have added some information about the vegetation of this habitat and differences along our transect in the method section (2.1: new paragraph on the study area).

Line 160: Why exclude them? I can understand for Diptera which are sometimes difficult to identify but Madeira Lepidoptera are all well known (or do you consider adults too mobile?) And I guess very few Amphipods and Isopods species are presented.

RESPONSE: We have clarified this in the manuscript. Lepidoptera, Isopoda and Amphipoda were not studied since we have no expertise for a fast identification and creation of morphospecies

Line 172: No control traps in perturbed areas?

RESPONSE: This study focused on remaining (semi-)natural habitats and indeed had no control traps in perturbed areas during the period of study. However, as now we mention in the discussion these traps were also very efficient in detecting rare endemic beetle species at low elevation exotic sites in Terceira island  [Tsafack, N., Fattorini, S., Boieiro, M., Rigal, F., Ros-Prieto, A., Ferreira, M.T. & Borges, P.A.V. (2021). The role of small lowland patches of exotic forests as refuges of rare endemic azorean arthropods. Diversity, 13(9): 443. DOI:10.3390/d13090443]  .

Line 187: Maybe list the species here? If there is not too much this information can be nice to better illustrate your sampling.

RESPONSE: Thank you for this suggestion. We have added a table in Supplementary material of all species recorded on the elevation transect in summer with parameters of their elevational range (mean, minimum, maximum) and abundance. We agree with you that this is useful information to add to this article.

Line 244: All Aranea are predators, while only a part of Coleoptera and Hemiptera, Psocoptera are not. Do you discuss this point later? This may partly explain your distribution results.

RESPONSE: Thank you for this interesting suggestion. We come back to this in the discussion, as this might indeed partly explain the diversity patterns along our elevation gradient. However, it is too much to also analyse the elevational diversity patterns of groups separated by diet in this study. This will be covered by a future study.

Figure 2 please use a more distinct color to separate native from uncertain even if only one dot is present.

RESPONSE: We changed the color

Figure 3 and figure 4: A little confused for me. A comparison by order may be missing (native Aranea + endemic versus introduced and the same for the other orders). Finally, a main point is to check whether native species (including endemic species) have a different distribution pattern from that introduced. The same goes for comparisons of incidence and abundance. Don't you think that by combining order + feeding type (i.e. predatory or not) + introduced / native (including endemic) you might not achieve a better resolution (I might be completely wrong, but can you check it)?

Beta-diversity results. Same thing… I think you have to compare the values ​​for native Aranea versus introduced Aranea and so on (for each order) to be able to compare your results. I'm not convinced that an overall comparison of introduced / native / endemic arthropods is very informative as your different guilds may have very different dispersal abilities depending on the diet.

RESPONSE: Thank you for these interesting suggestions. We have explored the patterns of  indigenous (native + endemic) species per order, as shown in the supplementary material (Figure S3g-h). However, we have found the most contrasting patterns of species richness for endemic and introduced species while native non-endemic species show a monotonic decrease of species richness with elevation in close resemblance to all species combined. This also applies to indigenous (native and endemic) species per order. We therefore emphasize the differences between endemic, native non-endemic and introduced species. Furthermore, the numbers of introduced species per order in this dataset were too low to allow robust separate analyses.
We have emphasized in the results for indigenous species per order in the discussion.

Furthermore, we agree that dispersal and dietary differences between species guilds might have important influences on distribution patterns. Suitable functional trait data is available, but this question will be addressed in a different future publication; including it into this paper would add too much complexity. We have raised this relevant question in the discussion.

Line 407 to 429. I broadly agree with your conclusions but as mentioned before you cannot conclude with precision without taking into account the rate of dispersal as a function of the order (and even among orders: non-flying beetles will not disperse. as much as the flying ones) and the guild type. Most of the time, predators have a larger territory than phytophagous. Perhaps to provide detailed examples of some selected species such as your dominant seedeater: Plinthisus minutissimus, which is a very wide range species.

Line 482 to 487: Again, these differences could be due to the dispersal abilities of the species. Low dispersal species tend to be governed by replacement of species, while mobile species are more dependent on changes in species diversity.

RESPONSE: As mentioned above, effects of dispersal differences between species will be subject of a different paper. Here, we have now included the potential influence of dispersal differences in the discussion.

Reviewer 4 Report

The MS “insects-1337768” deals with arthropod biodiversity on a mountainous system in Azores archipelago. The paper is well written, in a concise and correct English. I only have some minor revisions before final acceptance.

  • Line 14. “Arthropod species alpha and beta diversities change”. Better change in “change in alpha and beta diversity of arthropod species”.
  • Line 16. “SLAM traps”. Please avoid acronyms in the summary and the abstract.
  • Lines 44-46. There are some problems in the layout and formatting.
  • Lines 54-55. Given that it is a key topic, some more references are needed.
  • Line 137. It should be “we predict that”.
  • Materials and methods should start with a description of the study area linked to the map in Figure 1. Therefore, this part should not be included in the Introduction as it is now.
  • Lines 179-183. This sentence is unclear, maybe you mean “once every 90 days”? please, rephrase.
  • The rest of the methods is well described and replicable.
  • Results are clear and well shown.
  • Discussion: Authors should start with the meaning of their results and then move to comparison with other studies and the assessment of limitations. Therefore, I suggest authors to revise the structure of this chapter.

Author Response

REFEREE 4

The MS “insects-1337768” deals with arthropod biodiversity on a mountainous system in Azores archipelago. The paper is well written, in a concise and correct English. I only have some minor revisions before final acceptance.

RESPONSE: Thank you very much for your positive review. We have carefully taken your suggestions into account.

  • Line 14. “Arthropod species alpha and beta diversities change”. Better change in “change in alpha and beta diversity of arthropod species”.
  • RESPONSE: we revised this sentence
  • Line 16. “SLAM traps”. Please avoid acronyms in the summary and the abstract.
  • RESPONSE: Thanks for pointing this out, we corrected this.
  • Lines 44-46. There are some problems in the layout and formatting.
  • RESPONSE: Thank you for pointing this out
  • Lines 54-55. Given that it is a key topic, some more references are needed.
  • RESPONSE: You are right that extensive literature about this topic is available. Therefore we cite a review and chapter that provide an overview of scientific advances in this field. We have added two relevant and more recent references.
  • Line 137. It should be “we predict that”.
  • RESPONSE: this has been revised
  • Materials and methods should start with a description of the study area linked to the map in Figure 1. Therefore, this part should not be included in the Introduction as it is now.
  • RESPONSE: Thank you for noticing this. We have restructured this part and have added a new paragraph 2.1 ‘study area’.
  • Lines 179-183. This sentence is unclear, maybe you mean “once every 90 days”? please, rephrase.
  • RESPONSE: This is now included in the first rows of 2.2.
  • The rest of the methods is well described and replicable.
  • Results are clear and well shown.
  • RESPONSE: Thank you for these positive comments
  • Discussion: Authors should start with the meaning of their results and then move to comparison with other studies and the assessment of limitations. Therefore, I suggest authors to revise the structure of this chapter.
  • RESPONSE: Thank you for this remark. We have improved the structure in parts of the discussion. We choose to structure the discussion according to our research questions and hypotheses (1-3). For each question, we first explain the meaning of our results in relation to our hypothesis and then in comparison to other studies as you describe. Structural changes were made in paragraph 4.1 and 4.2 in particular.